# “We’re Still Struggling a Bit to Actually Figure Out What That Means for Government”: An Exploration of the Policy Capacity Required to Oversee Robot Technologies in Australia and New Zealand Care Services

**DOI:** 10.3390/ijerph19084696

**Published:** 2022-04-13

**Authors:** Helen Dickinson, Catherine Smith, Nicole Carey, Gemma Carey

**Affiliations:** 1Public Service Research Group, University of New South Wales, Canberra, ACT 2612, Australia; 2Melbourne Graduate School of Education, University of Melbourne, Parkville, VIC 3010, Australia; catherine.smith1@unimelb.edu.au (C.S.); gemma.carey@unsw.edu.au (G.C.); 3Wyss Institute of Biologically Inspired Engineering, Harvard University, Cambridge, MA 02138, USA; nic.carey@autodesk.com; 4Autodesk Robotics Lab, Autodesk Inc., San Rafael, CA 94903, USA; 5Centre for Social Impact, University of New South Wales, Sydney, NSW 2052, Australia

**Keywords:** robotics, care, policy capacity, Australia, New Zealand

## Abstract

Many countries are experiencing a “care crisis” driven by increasing demand for care services alongside difficulties in recruiting and retaining an appropriate care workforce. One of the solutions offered to this is the use of robotic technologies. While there are several positives produced by robots, they are not without challenges and have the potential to be misused. History shows disruptive technologies require appropriate policy capacity for these to be effectively stewarded so that we can secure the positive gains of these without encountering potential harms. In this paper, we explore the types of policy capacity needed to oversee robotic technologies. Drawing on interviews with 35 key stakeholders involved with the implementation of robots in Australian and New Zealand care services, we identify the capabilities required at the individual, organisational, and systemic levels across the analytical, operational, and political domains. We found the respondents perceived a lack of policy capacity to oversee robotics in the government. However, these gaps are less in respect to technological skills and abilities and more in respect to the system’s impacts and effects of these technologies. We conclude by outlining a summary of the capabilities required to oversee robots in complex care systems.

## 1. Introduction

An oft-heard refrain in the policy and academic literature is that we are facing a looming care crisis. Many countries are presently experiencing significant changes in relation to care services [1]—groups receiving these services are increasing in numbers, becoming older, have greater levels of disability and chronic illness, and have higher expectations about quality [2]. At the same time, care service providers are finding it increasingly difficult to recruit an appropriate workforce [3,4,5]. Technology advances offer a potential solution to these twin demand and supply-side pressures, particularly through the application of robotic technologies. Indeed, calls for the increased use of robotic technologies in health and social care services have intensified during the COVID-19 pandemic [6]. Unlike humans, robots cannot get sick and are less likely to transmit infection to vulnerable populations. 

Robots could plausibly fulfil several roles in our care services. They may replace or assist carers in manual tasks, such as lifting, can undertake routine tasks, such as taking blood pressure and oxygen levels, and remind people to take medications; they can also act as a source of social interaction for individuals who might be isolated [7,8]. In some areas, this could have clear positive impacts, delivering efficiencies and enhancing effectiveness, quality, and safety [9]. Yet, as the Australian Human Rights Commission [10] notes, “like any tool, technology can be used for good or ill… modern technology carries unprecedented potential on an individual and global scale. New technologies are already radically disrupting our social, governmental, and economic systems” (p. 7). New technologies are, therefore, disjunctive, offering significant advantages alongside the potential for misuse, including unintended consequences that need careful consideration to ensure such developments do not negatively impact particular groups. For example, a number of authors have written about ethical concerns with respect to the use of robotics in care services [11,12]. History shows that disruptive technologies require appropriate policy capacity for these to be effectively stewarded so that we can secure the positive gains of these without encountering potential harms [13]. However, as we have previously argued, at present, the market seems to be the main driver for the adoption of these technologies, and the government is not playing a strong role in steering these technologies in the case space [14]. As Wellstead and Stedman argue, “Ensuring strong policy capacity based within a public service is a critical factor in avoiding policy failure” [15] (p. 894). Therefore, there is a crucial need to be clear about the types of policy capacity required to ensure this is in place within appropriate agencies. This paper unpacks this by examining the policy capacity requirements to oversee the use of robots in care services.

Despite being an important component of policy success, policy capacity is often ill-defined: “While it is a cliché to argue having adequate policy capacity is a necessary pre-condition for policy success, there are many disagreements about the detailed conceptual and definitional aspects of the subject which have hindered efforts at better understanding and diagnosis, and improved practice” [16] (p. 166). At its most basic, policy capacity is the set of skills and resources—or competencies and capabilities—necessary to perform policy functions. Policy capacity is often described as existing at different levels (i.e., individual, organisational, and systemic) and having various dimensions (e.g., analytical, operational, and political). Following Wu et al. [16], Wellstead and Steadman [15] set out a framework of policy capability typologies (Table 1). 

In the growing literature on policy capacity, there are a number of studies that have sought to define and measure this in areas such as health [17,18], green industries [19], climate change [20,21], and even COVID-19 [22]. There is a gap in the literature with respect to robotic technologies and the types of policy capacity needed to oversee these. Without a clear articulation of this capacity, agencies lack the ability to verify the skills and capabilities required or to measure their progress against these. This paper adds to the extant literature by providing an analysis of what policy capacity is required to oversee robotic technologies. 

Drawing on interviews with 35 key stakeholders involved with the implementation of robots in Australian and New Zealand care services, we identify the capabilities required at the individual, organisational, and systemic levels across the analytical, operational, and political domains. We found that respondents suggested that at present, governments are not playing a strong role in overseeing these technologies, and there are gaps in terms of policy capacity and associated skills and capabilities. However, the gaps reported are less in relation to technological skills and are more associated with the broader system impacts and effects of the implementation of these technologies. In this paper, we map out the policy capacity required to oversee these types of technologies. Our argument is structured as follows. In Section 2, we provide an overview of the research approach before setting out our findings across the analytical, operational, and political domains. We conclude by setting out an overview of the policy capacity requirements to oversee robots in care services. 

## 2. Materials and Methods

The data presented come from a study exploring the implementation of robots in care services in Australia and New Zealand. This work sought to explore the roles that robots should and, even more critically, should not play in care delivery, and the role that the government has as a steward in shaping these roles. The project was afforded ethical approval by the University of New South Wales Human Research Ethics Committee (HC171025). A qualitative approach to research was adopted using semi-structured interviews [23]. Semi-structured interviews are typically used to gain a detailed picture of a respondent’s beliefs or perceptions of a particular topic area [24]. A purposive approach to sampling was adopted to identify interviewees [25]. We sought to engage a range of experts across Australia and New Zealand with roles in policy (at different levels and different areas of care provision), provision of care services, academics and other expert commentators in the topic area, and technology suppliers. All interviewees had experience in the use of robotic technologies in care services within their respective jurisdictions. 

In total, 35 interviews were conducted with these stakeholders (see Table 2). Interviews lasted between 30 min and 90 min and were recorded and transcribed verbatim. After Kallio et al. [26], we developed an interview schedule that covered issues such as where robots are currently being used; the advantages and disadvantages of robots in the sector; considerations when introducing robots; future roles for robots; the role of the government in overseeing robots. Although the respondents were not asked explicitly to define care, further analysis was undertaken to identify the ways that interviewees talked about care and the implications of this in developing a definition of care. The data were analysed using a thematic approach [27] in NVivo. “Like” data were grouped together to form categories and subcategories. These categories were developed into more substantive themes by linking and drawing connections between the initial categories and hypothesising about the consequences and likely explanations for the appearance of certain phenomena [28]. 

In the findings that follow, quotes from interviews are reported to illustrate key points and themes. In accordance with our ethical approval, the quotes are not ascribed to individuals and are simply labelled according to the country that the individual primarily works in (e.g., AU01, NZ03). 

## 3. Findings

In this section, we provide an overview of our findings, setting these out against the analytical framework of operational and political categories as elucidated by Wu et al. [16] and further elaborated on by Wellstead and Steadman [15]. Before we explore the key themes in more detail, we briefly reflect on the state of capacity to oversee robotic technologies broadly. The interviewees generally expressed the belief that governments lack the capacity to do this effectively. For example, one respondent described the following:

“I feel like with robotics and AI, everyone keeps saying we want to be on the front foot but we’re still struggling a bit to actually figure out what that means for government and exactly where we need to be playing and not playing and where the gaps are that we need to be jumping into or supporting or helping so I think we’re still trying to figure that out”.(AU07)

The speed with which these technologies are being developed and rolled out poses difficulties for governments as they attempt to keep pace with this and to keep their skills, capabilities, and capacities in line with this. One of our interviewees reflected on this and the speed that policy typically operates at: 

“I think one of the really interesting things about where you’re, the space that you’re looking in, is to me, technology flies. It’s super, super fast, but policy is very slow. So, how do we effect change in policy that’s going to keep up with the change in technology?”.(AU09)

If agencies are to develop policy capacity, we need clarity over what this comprises precisely and the types of skills and capabilities that are required for this. In the following sections, we map out these requirements. 

### 3.1. Analytical

Several interviewees commented that governments often lack skills in relation to technology and the various implications that disruptive technologies can hold in terms of things such as thinking through ethics, data storage, and issues relating to privacy. As a result of these deficits, some interviewees expressed the belief that governments have some significant gaps in terms of their ability to design and implement effective policies that involve robotic technologies. A number of respondents felt policymakers lack some basics around what different technologies are that policies are being developed around and the metrics that are being used to assess whether these are effective. As they described,

“I mean I would say step number one in thinking about what does policy making look like in this space is being really clear about what we imagine we are making policy around. I think that means getting down to terms and then I think the second piece there is well you know, having defined robotic objects, what is it that we want to use as the evaluative mechanism of their virtual value? [Laughs] Like you know, are we talking about classic automation? So, it’s about time and labour saved, so productivity and efficiency gains. Is it about an aged care facility’s—an absence of labour? Is it about quality-of-life stuff? Is it around energy efficiency stuff?”.(AU04)

This interviewee believed that without a clear sense of what these technologies are and the impacts they will have, it is difficult to develop effective policy. Therefore, there needs to be some understanding of the basic concepts and models so technologies can be differentiated from one another and their intended impacts can be understood. 

There was some debate within the interviews as to whether it is appropriate for governments to hold actual technical skills and expertise with respect to these technologies, particularly given the speed with which they change and develop. Rather than develop these capabilities across public service organisations as a whole, some governments were instead recruiting individuals who would be the expert in such technological knowledge. As one interviewee described, 

“Governments are starting to… have some kind of technology expert. Often, we talk about a chief technology officer. If you’ve got technology in there, you should have a chief technology officer who is responsible to make sure that something happens. It might be that all the major government departments have their own CTO [Chief Technology Officer]. They coordinate. There has always been technology, but if it’s becoming more important for that decision making, you could say you want to have a chief technology officer. It might be a part time person. But they should be a technology expert. But you’ve got to have somebody who is on your side of the fence advising you for your organisation”.(NZ05)

In this case, it is the job of a key individual to retain the technological expertise and to use this to advise colleagues around any decisions that may be taken with respect to the implications of this. Without having this expertise in-house, governments rely on others, and some may have a conflict of interest—for example, where a provider wants to sell technology to the government. The informational asymmetries that arise in this situation were seen as problematic to some. 

Others saw less need for governments to have technical expertise in all areas but instead saw the need to be able to think through relationships with technologies. One interviewee explained this by providing an example of how some schools are treating technology in different ways:

“Schools have really different approaches to how they introduce kids to technology. Some of them are like well what we’ll do is we’ll teach you coding skills, or we’ll teach various different technological skills, which I’m a bit dubious about in some senses. Other schools have really said well actually what’s important in terms of new technologies is working with kids to think about the role that technologies do and will play in their lives and thinking about their relationship to technologies”.(AU15)

This interviewee went on to explain that significant investment in building the technological capabilities of individuals would be inefficient in the long term given the speed with which these skills change. From their perspective, what was more important was to make public servants technologically literate in thinking through how technologies such as robotics might play a role in the lives of individuals and communities and some of the types of questions that this might raise. 

In the absence of generalised technological expertise in robotics or AI, a number of interviewees suggested what was important for individuals was an ability to partner with a range of individuals and organisations with an interest in this space. Robotics and their use in public services were seen by most as a collaborative effort that requires the input of multiple organisations and disciplines if it is to be effective. As one interviewee explained, *“I don’t think a designer should have, doesn’t matter how great, should have the power to make all these decisions. Not the AI expert. Not the social scientist alone. You need everyone”* (AU15). The implication of this in terms of policy capacity is that what governments need is the ability to effectively communicate across the policy sphere and to understand the various positions and drivers of different partners. We will come back to this again in the next section, as there are organisational implications for this, but at an individual level, this requires capabilities around communication, as one interviewee describes: 

“From a government’s perspective we’re not aware of a lot of this stuff unless we have these conversations and the challenges and barriers in this space…. I think we, at least from us from a bureaucrat space, we know we need ethics frameworks in place and—but to me it’s a real partnership with the private sector, with the universities, with everyone working together because we know very little in this space and it’s not until we actually have these conversation… We’re not the experts and we are not the experts in any of this area but if there is regulation that needs to be removed and things that we need to be putting in place then we need to be having these conversations and I don’t think we’re great at being in those conversations”.(AU07)

As this individual identified, they are aware there are implications of these technologies in terms of ethics and regulation, but it is not possible to design and implement effective policy without actively engaging with a range of stakeholders across the system. This requires some basic skills and abilities in collaboration to achieve. 

### 3.2. Operational

At the level of the organisation, most of the skills and capabilities that interviewees viewed as important relate to overseeing the market, helping providers to interpret the evidence base, and providing an appropriate regulatory framework. 

Some interviewees were concerned that markets might evolve that are not good for consumers as they do not promote competitive behaviours. As they explained, “*Anti-competitive product ecologies. Of the sort that you’ve seen with like iTunes or with the Google Play Store. So you can imagine that these robots only work if you have the other robots in the house are all—so you’re locking people into a product ecology in ways that I think is probably bad*” (AU11). Given that these types of issues have emerged around digital markets, several respondents were mindful of these and saw a role for governments in preventing the exploitation of robotics consumers. Another facet of market management is providing funding, and a few interviewees felt that governments need to be able to identify where gaps exist in markets and provide funding to help fill these. For example, 

“I think that that role of governments in terms of whether there’s gaps in funding, as well like the barriers, the things that governments gets involved with but if we think that the market isn’t going to drive development in areas but we think there’s really good work to be done with AI, I think that will be interesting to explore whether we think we could—there’s opportunity to move a lot more quickly in some of those areas that may not be commercially viable”.(AU07)

In this case, the role of public servants is to identify potential areas that may not be well served by the market and find ways to encourage providers into these. 

Several interviewees felt that at present, the evidence base around robotic technologies is unclear, and providers of services (e.g., health, aged care) may not understand the nuance of this and, therefore, whether these technologies are effective. As such, one important role for governments was identified as being able to understand and interpret the evidence base for partners. As one interviewee explained, 

“We need to understand capabilities, so versions, capabilities of actual products. If aged care settings go and invest public money or foundation money or whatever money, it’s really up to them to make the thing work for them as best they can. Always what I try to say with the organisations that I work with: they should be working … with researchers and developers. Developers need to be working collaboratively with researchers and the public. That, to me, is the key, because there’s so much crap out there”.(AU14)

As this quote illustrates, without this level of knowledge and understanding, there is a risk that some providers who are funded by public money might spend their resources on technologies that do not deliver the anticipated benefits. This interviewee went on to explain that this is an issue with technologies such as robotics, which have a high degree of interest from the public. They told us “*That also worries me that people write—people, researchers write papers that don’t have very good methods then publish. Then media get told of it or someone—then it goes in this sort of circle*” (AU14). Their perspective—and this is one shared by several other interviewees—is that there is insufficient quality research into robotics, and their benefits are often hyped in the media. A key role for governments is, therefore, in the evaluation and assessment of robotics and to share these findings in an accessible way for potential purchasers of these products.

Other interviewees noted these evaluation and assessment skills are even more important because so few of these technologies are developed in Australia. This means that these technologies have a series of assumptions built into them that may not be applicable to the Australian context. One interviewee explained this as follows: 

“I think maybe a good place for policy to start looking is around policies that allow, or frameworks that allow those things to be evaluated and assessed for use in the Australian context, so maybe that’s a starting point, and whether or not—if we can get those frameworks into place that will allow us to evaluate different technologies around, then we can streamline the implementation into our own systems here. Maybe that’s a place where policy can be effective in the first instance, rather than running and trying to catch up”.(AU09)

Regulation is an area that most interviewees spoke about as being an important area for governments to have skills and abilities. Most interviewees believed that standards should be developed for robots, although there was less certainty regarding what these might look like or comprise. As one interviewee explained, 

“I think we’re talking about regulatory standards. A lot of people want to think about that. I think it’s very hard, because we don’t know and it’s not clear what the robots are going to be doing when they’re doing things with people. You’re predicting a little bit. What you could do is you could look at your current—if it’s a medical device, there is a bunch of standards you have to meet…there is a bunch of regulation around heavy machinery, factory automation and how you have to protect people in factories. We are just not used to it in homes. People—I tell people you should look at all the other standards before you start inventing something new, because there is a lot of it already there. But then look at that and see what it’s not doing”.(NZ05)

This interviewee went on to describe the Japanese system where there are clear standards about what robots can and cannot do and the features that they are able to possess. A number of interviewees cited other countries that have processes in place in order to assign a quality guarantee to particular technologies for certain purposes. As another interviewee reminded us, many of these technologies are constituted by a series of assumptions and expectations that are built into these technologies and these have important implications for how they are used:

“They [robots] are in fact culturally and constituted and driven by medical insurance capacities, so knowing—sitting inside those robotics objects, what mechanisms are built in there that are based on medical data, that are based on expectations about human activity? How do we get to scrutinise those?”.(AU04)

As this interviewee illustrated, one of the challenges, particularly where AI is built into robots, is that the algorithms that drive the decision making in these technologies are not always transparent. In some cases, this is because they are seen as proprietary by their creators. 

One of the areas of particular concern in terms of regulation is in relation to data and who they are shared with and why. One of the facets of these technologies is that they create new types of data and potentially around the most “vulnerable” of groups. Most of our interviewees expressed the opinion that those individuals who live in congregate care settings (e.g., aged care facilities, group homes for people with disability) required appropriate protection in terms of their privacy: 

“Let’s be clear. If you were to install stuff in your home, you have a different expectation of your privacy in your own home than you do if you’re in a nursing home and the bar is actually higher on an elder care facility. So you install that stuff in your home and you inadvertently bring hackers in. Well that’s not—okay, bad. You sign into a facility where they are agreeing to care for you, them violating your privacy is a completely different deal and it ought to be much worse”.(AU04)

One interviewee saw this as a relatively straightforward task, as the following quote explains:

“I think the information privacy kind of thing needs to be dealt with data analysts and data informatics people. They’ll solve that. I mean that’s not a policy—the only issue for policy there is that they need to keep up to speed with what’s happening in data security and embed that into their thing. I don’t see that as something that they need to reinvent the wheel on”.(AU07)

According to this perspective, data security and privacy are issues that appropriately skilled data analysts should be able to deal with, but for others, the debate is more nuanced. As one interviewee explained,

“It’s not just how do you get people to understand the technological infrastructure, it’s how do you get them to understand that the technology is just the beginning and what they are in fact doing is creating a new class of material that now needs to be protected. So, I don’t want to think about it as data because I don’t think it is just data because that data also primes algorithms, it’s images, it’s all this stuff. So effectively you are now creating a new class of information that has never existed before and how do we want to think about that class of information as being something that needs to be protected or trafficked in”.(AU04)

This interviewee was arguing that privacy is important to think about because the data and their use, in terms of feeding algorithms and being shared with technology providers, means that it is of a different nature to the sorts of data and privacy issues that have preceded it. This individual went on to describe the potentially negative impacts of this data collection: “*I’ve always thought that most of that technology is really good if you’re a batterer or a purveyor of domestic violence, we’ve just created the perfect technology for you. You know exactly where she is all the time*” (AU04). Given the very real potential impact of such sharing of data, care needs to be taken to ensure that systems are set up appropriately and cannot be used in unintended ways.

A further issue in terms of regulating data is posed by the international systems that sit around many technologies. What this means is that data are not assured to remain onshore, as one interviewee explained: “*The problem in Australia is that a lot of that data goes overseas. I think that’s a problem and I think maybe we need to move to having that onshore rather than offshore*” (AU09). Data that are stored overseas will be subject to different rules than that which stays within Australia. Regulators, therefore, need to be attuned to understanding what the implications of this are in terms of the rights of consumers and organisations. 

### 3.3. Political 

At the political level, the types of factors discussed tend towards strategic leadership activities across a number of domains such as workforce, education, and infrastructure, but more fundamentally around what robots should and should not do and how these technologies will be integrated into socio-technical systems. In doing so, many interviewees highlighted they viewed the introduction of some robots in health and care systems as interventions in complex systems. Several interviewees spoke about the ripple effects of introducing some kinds of robotic technologies, particularly when considered over the longer term. As one interviewee explained, 

“There’s a lot of secondary and tertiary effects associated with this stuff. A lot of it is things that we don’t foresee. It’s like invasive species. You can think you’re doing a good job when you’re adding it, but you don’t really understand. That’s why we need more people thinking about what the secondary and tertiary consequences are”.(AU04)

As this interviewee reflected, new technologies can have a range of impacts that go beyond the function they have been designed to deliver. It is, therefore, important that a systems perspective is taken on these technologies and the potential broader implications are considered. This sentiment was reiterated by another interviewee, this time in terms of the costs of these technologies:

“For every one dollar of technology invested, there’s a corresponding nine dollars invested in things like staff education, systems redesign and all of that”.(AU01)

Introducing robotic technologies can have implications for various other whole areas of government issues, such as security, data sovereignty, infrastructure, workforce, education, and market stewardship. To this extent, the introduction of disruptive technologies, such as robotics, needs some careful foresight work, as they have the potential to create ripple effects across governments and society. 

In addition to having the skills and abilities to support thinking through the broader implications of technological change, a number of interviewees expressed the belief that the government needs to take a strong thought leadership role. Such a role would identify where the boundaries are with these types of technologies, where people do or do not want to see these, and where the tolerance of risk lies. As one interviewee explained, *“So I do think it has to be there, conversation at the general level, public level I think. You know the general population needs to be involved in the conversations” (AU13).* Again, given the fast pace with which some of these technologies are developing, this was seen as an important action to ensure that the community is happy with their use:

“Well, it’s beyond the conversation phase now, because the technology is already outpacing our ability to regulate and legislate for it, so we’re way behind. The real question is, what are we going to allow—are we just going to be a big experiment, where all the stuff is thrown upon us and we see what happens? Then just say, oops, sorry if that was the wrong answer. Or are we going to then end up overreacting and throw the baby out with the bath water and there was good there but now it’s—we can’t use that because all of it’s dangerous?”.(AU08)

As we have already indicated, several interviewees highlighted significant ethical issues associated with technologies. Some went further than this and observed that there is the potential for these technologies to be used in ways that enhance inequities. As one respondent described, “*Then, I suppose there is the digital divide. We’re very conscious already that if you’re well-educated, have a pretty good income, you can buy the latest technology, you can be on the information superhighway, or whatever it’s called*” (NZ05). As other respondents noted, this is not to say that robots inherently develop inequities, but if not carefully considered, they may exacerbate divides across, for example, generations, socio-economic groups, or levels of education. As one interviewee explained, *“Robots are no different than any other technological wave that have the possibility for reinforcing remarkable social divides. Of course they do”* (AU04).

As set out in the previous sections, many interviewees commented that at present, most robotic technologies are developed in other countries and are imported for use in Australia. This has implications in terms of the assumptions that are built into these devices and potentially where data are stored and how they are used. As such, some interviewees expressed the belief that the government should engage in activities to try and ensure that more of these technologies are built in Australia. For example, 

“Like I really think that we need to start bringing in really good quality standards and developing a lot of the stuff onshore. At the moment, the stuff that we’re getting from overseas is fine, it works, there’s no problem with it. But there’s a cost associated with that. Can we do it better, cheaper here? I don’t know. But if those people—if we’re going back to that manufacturing thing and the point you made earlier, if we’re taking people off car manufacturing and putting them on robot manufacturing, they’ve still got a job. So maybe we need to water the grass on our shores if we want a competitive sustainable foot in the robot door”.(AU09)

As this interviewee explained, when we think of robotics in terms of a broader system, it may not be cheaper to simply import these technologies. If these are to be used in mainstream services, then there may be jobs that can be created in their manufacturing, particularly when considering other manufacturing roles that have either gone in recent years (e.g., cars) or have been significantly reduced. Some of those we spoke to saw robotics as such a growth area nationally and internationally, and they strongly believe that Australia needs to invest in this area or else this will pose additional challenges in the future. Many interviewees felt that at present, there is insufficient investment in innovation activities in terms of the money that the government is willing to commit, but also in terms of working to bring together a range of the different stakeholders that have a role to play in this area:

“I think there needs to be coordinated approach in terms of the importance of investment in innovation, and linking industry, community providers or whatever, and also researchers together. Currently, we have to get—to do things like partnership grants, et cetera we have to get money from industry. Often industry don’t have that money, so I don’t know; it’s a problem”.(AU14)

Many robotic technologies are dependent on having appropriate infrastructure operating effectively, such as sufficiently fast internet connections. Without preparing this basic infrastructure, organisations will not be able to use more sophisticated technologies, as one respondent explained: 

“We need a faster Broadband … It’s a real problem in Australia though, because we’ve got—we don’t have—Broadband is not good. NBN [National Broadband Network] is not good. Really, it’s quite a challenge; we have houses that were built in the seventies, eighties that are double brick that you can’t even get a mobile device in. There is certainly still a lot of work to be done in that area”.(AU14)

A further resource frequently mentioned by interviewees was the workforce. Many interviewees noted the impact that robots might have on the types of work that humans undertake and this, in turn, has implications for the types of skills and abilities needed in the future. As one interviewee described, an area of concern for their area of government is “*the impact on the workforce, and how it might change demand for skills, therefore how we prepare workers through the education and training system for those kinds of roles*” (NZ05). Such an observation, therefore, has implications in terms of thinking through the types of skills, knowledge, and abilities that schools and training institutions need to be developing in students. 

## 4. Discussion

Our previous research demonstrated that Australian and New Zealand governments play a relatively limited role in overseeing the use of robotic technologies in care services at present [14]. Most care providers who use robots told us they had been most influenced by the manufacturers of these products, suggesting the market is leading the way in terms of the adoption of these innovations. This was seen as problematic, as there is potentially significant informational asymmetry and investments in these technologies may either not produce the impacts that care providers seek but may also pose other challenges or threats. In this paper, we built on this work to explore the issue of policy capacity specifically. As our findings show, the opinions on the capacity of governments to do this work were low, and several gaps were outlined by stakeholders. Moreover, many respondents expressed the belief that this situation is problematic, and should the use of robotic technologies significantly expand, they believe that this might raise a series of challenges for users of these technologies more broadly in areas such as efficiency, privacy, ethics, and the equity of outcomes, amongst others. In this paper, we have sought to outline the different skills, abilities, and knowledge that are needed to be developed to enhance this policy capacity. In Table 3, we summarise the various components of policy capacity outlined in the findings. This provides additional value to Wellstead and Steadman’s [10] framework, in the sense that it takes this broad background and elaborates the specific requirements for overseeing robotic technologies. This will be of interest to those public servants working in this area and fills an existing gap in the knowledge base. 

Robots are a technological intervention, but as our findings show, in terms of the skills, capabilities, and knowledge required to oversee these, there are actually relatively few “technical” skills outlined. Far more of the capabilities outlined by the interviewees are arguably more generic skills applied to this area. In terms of the individual level, there is some requirement for an understanding of different technologies and the parameters these might be assessed against. Moreover, many organisations saw the need for a Chief Technology Officer, or similar, who might have some specialist knowledge; however, beyond a relatively low level, the interviewees agreed that advanced technical knowledge was not needed across the organisation. Instead, this would be accessed through collaboration with a range of relevant stakeholders across several organisations. Our findings here accord with observations made by Pasquale [29], who wrote of his work that it “is to warn policymakers away from framing controversies in AI and robotics as part of a blandly general ‘technology policy’, and toward deep engagement with domain experts charged with protecting important values in well-established fields” (pp. 15, 16). The types of issues that were seen as more important and where most respondents believed there are currently gaps in organisations are in relation to issues around data collection, privacy, and ethics. Robotic technologies, particularly those powered by artificial intelligence, were widely seen to be creating new types of data that organisations have not had to previously deal with and that these developed new types of skills and abilities to address.

Many respondents viewed the use of robotic technologies as less of a case of straightforward technology adoption and more as an intervention in a complex system. One of the areas in which the respondents felt governments do lack knowledge and insight are the various models of care and what the implications of adopting robotic technologies might be for these. In the care systems that we studied, we have seen a gradual movement of the delivery aspects of these services from the government to third-party contractors [30]. An implication of this is that public servants might not have good insight into these services and how they operate on a day-to-day basis, raising challenges in terms of understanding the implications involved in making changes to these systems and services. There were, therefore, concerns that there are not simply informational asymmetries in terms of technologies but also care services more broadly. 

As we have outlined in our findings, there is an urgent need for governments to possess appropriate policy capacity if we are to harness the best of these technologies and guard against some of the potential negatives of these. Yet, over the past four decades, many studies of policy capacity have suggested that instead of improving and enhancing this, we have seen a general decline [31]. In Australia, for example, we have seen successive reviews of the Australian Public Service, including the most recent Independent Review [32], that have identified policy capacity as being a significant area of concern and which has not improved despite these reviews and programs to enhance it. Commentators feel there have been only small gains in some areas and even a decrease in policy capacity in some areas of the Australian Public Services [33]. Rather than developing areas that will be important to oversee these sorts of technological developments, we have seen reductions associated with a more general “hollowing out” of the public service [34,35] as policy capacity is reduced in the public service and is replaced through the use of consultants and other third-party agents. What is clear from our research is that if governments are to effectively oversee robotic technologies, policy capacity is an area that will require significant investment. Relying on outside agents to deliver these aspects of policy capacity will create a number of system impacts that may be detrimental. 

## 5. Conclusions

Governments need appropriate policy capacity if they are to oversee the implementation of policy. Yet, there is somewhat of a gap in the literature relating to the types of policy capacity needed to oversee robot technologies in the care space. In this paper, we undertook semi-structured interviews to explore the types of policy capacity and associated skills and capabilities needed to oversee robots in care services. The respondents suggest that there are some important gaps in the current care systems, which are largely being led, at present, by the providers of technologies. In this paper, we outline the policy capacity required, noting that the respondents put rather little emphasis on technological skills and were instead far more focused on more generic abilities associated with collaboration, foresight, regulation, and market management. These capabilities are all related to the idea that the oversight of these technologies is less a case of straightforward technology adoption and more of an intervention in a complex system that raises a series of important implications across sectors, organisations, and institutions.

## Figures and Tables

**Table 1 ijerph-19-04696-t001:** Wellstead and Steadman’s framework of policy capacity typologies.

Level Dimension	INDIVIDUAL	ORGANISATIONAL	SYSTEMIC
*Analytical*	**Analytical Capacity** Knowledge of policy substance, analytical techniques, and communication skills	**Technical Capacity** Data collection; availability of software and hardware for analysis and evaluation; storage and dissemination of operational information; e-services.	**Knowledge System Capacity** Availability and sharing of data for policy research and analysis; availability, quality and the level of competition of policy advisory services in and outside of government; presence of high-quality educational and training institutions and opportunities for knowledge generation, mobilisation, and use access to information
*Managerial/Operation*	**Managerial Capacity** Strategic management, leadership, communication, negotiation and conflict resolution, financial management, and budgeting	**Administrative Capacity** Funding, staffing, levels of intra- and inter-agency communication, consultation, and coordination	**Governance capacity** Levels of inter-organisational trust and communication; adequate fiscal system to fund programs and projects
*Political*	**Political Acumen Capacity** Understanding the needs and positions of different stakeholders; judgement of political feasibility; communication skills	**Political Resource Capacity** Access to key policy-makers; effective civil service bargain; politicians’ support for the agency programmes and projects	**Legitimation Capacity** Level of public participation in policy process; public trust; presence of rules of law and transparent adjudicative system

**Table 2 ijerph-19-04696-t002:** Interviewees by background.

Organisation	Number
Academic expert/expert commentator	12
Provider of care services	5
Government department/agency	13
Supplier of technology	5

**Table 3 ijerph-19-04696-t003:** Policy capacity requirements to oversee robotics.

	Individual	Organisational	Systemic
**Analytic**	Ability to differentiate between different types of technologies. Ability to understand and assess different types of impacts in complex systems.	Specialist technological insight and understanding of robots and associated technologies. Understanding of different models of care. Understanding of appropriate regulatory arrangements and ability to develop these.	Horizon scanning and foresight. Ability to understand and analyse different evidence sources and communicate to partners. Ability to understand complex systems and the impacts that robotics might create across a broad range of fields.
**Managerial**	Strategic management Communication skills Ability to negotiate amongst diverse stakeholder groups	Ability to effectively collaborate with a range of stakeholders. Ability to analyse and disseminate evidence to a range of stakeholders. Ability to develop and communicate required care standards.	Understand the market and the different mechanisms available to steer this. Adequate fiscal system to fund innovations and stimulate market in underserved areas. Understanding of issues relating to privacy and data sharing and mechanisms involved to shape this. Ensure availability of infrastructure (e.g., broadband, mobile phone and data networks).
**Political**	Understand needs and positions of different stakeholders. Communication skills	Access to key stakeholders across a range of partners.	Strategic leadership across the system. Ability to understand implications of developments for the workforce and to work with partners to plan for this. Level of trust from the broad community to lead conversation about where and for what purposes robots should and should not be used.

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
