# Peer review of "“We’re Still Struggling a Bit to Actually Figure Out What That Means for Government”: An Exploration of the Policy Capacity Required to Oversee Robot Technologies in Australia and New Zealand Care Services"

_ijerph, 2022, doi:10.3390/ijerph19084696_

Round 1

Reviewer 1 Report

We All can see the care crisis during the COVID-19 situation. This is a good topic, and I hope if the writer explain what is the limitation of having this technology? 

Author Response

Thank you for your review. We have included a few references in the introduction that point to some of the challenges of using robots in care services.

Reviewer 2 Report

The authors present a qualitative study about the adoption of automation technology, and specifically robots, in the field of health services. The paper identifies and discusses the key themes emerging from interviews with key stakeholders. Finally, the paper provides an overview of the policy capacity requirements to oversee robots in care services mapped over Wellstead and Steadman’s framework of policy capacity typologies.

In this regard, the authors could better highlight the added value provided by their work in contrast with the state of the art (Wellstead and Steadman’s).

Author Response

Thank you for your review.  With respect to your comment about the added value of our work compared to the work of Wellstead and Steadman, we  take this broad framework and elaborats on this with respect to the policy capacity required to oversee robotic technologies specifically.  We have included an additional sentence to this effect within section 4.

Reviewer 3 Report

This article provides an assessment of robotic care based on extensive interviews. These interviews were designed to evaluate and explore the types of political abilities and related skills and abilities needed to supervise robots in care services. In my opinion, the discussed topic is justified, however, there are no specific examples of how to improve care using robots. There are also no proposals how it should be implemented. Merely drawing the attention of politicians to this problem is not enough.

Author Response

Thank you for your review.  In our background section we have expanded on the activities that robots might undertake in care services and how they improve care.  With respect to the point about implementation, we are not entirely sure what is meant here.  The paper is not intended to just draw the issues to the attention of politicians.  Our research finds that public servants lack policy capacity and the intention of this paper is to set out the kinds of skills and abilities that would be required to operationalise this.  Therefore the intention of the paper is that it should inform implementation. We have attempted to finesse this line of argument in the revisions.    

Reviewer 4 Report

The title of the article is more suitable for a newspaper article or an article in a news magazine or online resource. For a scientific article, it is better to use a title that indicates the problem being solved and, possibly, a method for solving it. It is recommended to bring the title in line with the tradition of titles for scientific articles in a scientific journal.

In the introduction, as a rule, the authors prove the relevance of the research using links to external sources. At this point in the article, it is inappropriate to give links to your own publications, since this does not at all prove that the research topic is relevant, they only prove that the authors themselves are working on this topic. Therefore, references to publications [1], [6] in the introduction are inappropriate. Publication [2] is a publication of a co-author of publication [1], that is, it is also not fully an “external publication”, in this sense it is also not entirely appropriate. Publications [3], [6], [7], [27] are not formatted according to the rules for citing, so it is impossible to find them, and it is not clear whether such publications exist. If these are booklets available on the Internet, then a link to the site should be given, for example, https://apo.org.au/sites/default/files/resource-files/2018-06/apo-nid176691_1.pdf. If these are booklets issued in paper form as hardcopies and do not have an Internet source, then you need to give a link to the publisher, city and country of publication. As a rule, scientific articles have authors, and they should be mentioned in such references. The reference of the authors to their own sources is allowed in those cases when it is necessary to rely on the results previously obtained by the authors in order to more clearly describe new results, and not just to report that research is being carried out in this direction.

The sentence in lines 50-52 is knocked out of the general context of the article. First, self-referencing by authors with the word "Authors" in square brackets is not accepted in scientific articles, at least not in technical articles. Secondly, the phrase “as we have already stated” is not good for a scientific article where brevity is required, therefore it is not necessary to repeat one’s own statements, this is not a university report or a monograph. Thirdly, it sounds very doubtful and ambiguous to refer to one's own earlier statement in that part of the article before which there were no own statements of the authors, because if you carefully look at the entire text from line 28 to this place, all statements in it are provided with links to publications, even if they were publications of one of the authors, in this case it is necessary to give a link to such publications, and not write "authors". Fourthly, this statement is very doubtful, since governments can still influence the acceleration of the development of certain technologies through priority funding of scientific research and (or) implementation in this area. Fifth, this phrase also occurs later in lines 70-81, and the same statement is already given as the result of global research, while in lines 50-52 this statement is given as an a priori statement by the authors of this article. In addition, this dubious phrase is repeated in the conclusion, in lines 453-456, again with the same ridiculous reference of the authors to themselves in the form "Authors". This is even more ridiculous because the conclusions on this article begin with a line about the conclusions that were given twice at the beginning of the article as a priori information from these authors themselves (and this information is very doubtful).

Table 1 appears to have been borrowed from [10], as follows from this article. It is hardly appropriate to completely reproduce the table from someone else's article.

The whole article is not scientific, it's just a selection of some interviews, perhaps from random people. But even if the interviews were taken from professionals, the answers show that there is nothing valuable and new in these interviews. The results of the interview are not even processed properly, all the meaningful information in the conclusion consists in the same dubious phrase that was previously unreasonably given in the introduction, and about which the authors themselves write that they had said it before.

The article does not contain anything useful new, informative.

It is impossible to improve an article by editing.

I recommend that you unequivocally reject the article.

Author Response

Thank you for your review.  In this we outline how we have responded to your comments.  We have included your comments and then our response appears below this in italics.  

The title of the article is more suitable for a newspaper article or an article in a news magazine or online resource. For a scientific article, it is better to use a title that indicates the problem being solved and, possibly, a method for solving it. It is recommended to bring the title in line with the tradition of titles for scientific articles in a scientific journal. 

This is a qualitative study based on interview data.  As such it is not uncommon to use a quote within the title of the paper.  We do not think the title is not in line with traditional titles and the other 3 reviewers do not cite this as a problem.  We defer to the editor for further guidance on this issue. 

In the introduction, as a rule, the authors prove the relevance of the research using links to external sources. At this point in the article, it is inappropriate to give links to your own publications, since this does not at all prove that the research topic is relevant, they only prove that the authors themselves are working on this topic. Therefore, references to publications [1], [6] in the introduction are inappropriate. Publication [2] is a publication of a co-author of publication [1], that is, it is also not fully an “external publication”, in this sense it is also not entirely appropriate.

We believe we have used a number of sources to demonstrate the relevance of the work.  One of the challenges, as we argue in the paper, is that very few researchers in our field have examined this issue to date.  There has been no study of policy capacity and robotics in the literature.  We build on the literature on policy capacity (of which none of our teams are authors) to outline the relevance and importance of this concept.  We then provide some background to the issue of robotics with respect to the public management literature, of which we are the primary authors in that field.  In our previous work we have found that governments are not proactive with respect to this area.  We build on this observation to provide an original contribution to knowledge.  This is not an issue any of the other reviewers have picked up and we defer to the editors as to the appropriateness of this approach. 

Publications [3], [6], [7], [27] are not formatted according to the rules for citing, so it is impossible to find them, and it is not clear whether such publications exist. If these are booklets available on the Internet, then a link to the site should be given, for example, https://apo.org.au/sites/default/files/resource-files/2018-06/apo-nid176691_1.pdf. If these are booklets issued in paper form as hardcopies and do not have an Internet source, then you need to give a link to the publisher, city and country of publication. As a rule, scientific articles have authors, and they should be mentioned in such references. The reference of the authors to their own sources is allowed in those cases when it is necessary to rely on the results previously obtained by the authors in order to more clearly describe new results, and not just to report that research is being carried out in this direction.

As a study on policy it is not unusual to refer to policy publications.  The ones referred to are available online and in hard copy as publications that have been produced by government bodies.  As far as we are aware they are appropriately referenced according to the journal guidelines, but again are happy to defer to the editors if there are issues with these particular sources. 

The sentence in lines 50-52 is knocked out of the general context of the article. First, self-referencing by authors with the word "Authors" in square brackets is not accepted in scientific articles, at least not in technical articles. Secondly, the phrase “as we have already stated” is not good for a scientific article where brevity is required, therefore it is not necessary to repeat one’s own statements, this is not a university report or a monograph. Thirdly, it sounds very doubtful and ambiguous to refer to one's own earlier statement in that part of the article before which there were no own statements of the authors, because if you carefully look at the entire text from line 28 to this place, all statements in it are provided with links to publications, even if they were publications of one of the authors, in this case it is necessary to give a link to such publications, and not write "authors". Fourthly, this statement is very doubtful, since governments can still influence the acceleration of the development of certain technologies through priority funding of scientific research and (or) implementation in this area. Fifth, this phrase also occurs later in lines 70-81, and the same statement is already given as the result of global research, while in lines 50-52 this statement is given as an a priori statement by the authors of this article. In addition, this dubious phrase is repeated in the conclusion, in lines 453-456, again with the same ridiculous reference of the authors to themselves in the form "Authors". This is even more ridiculous because the conclusions on this article begin with a line about the conclusions that were given twice at the beginning of the article as a priori information from these authors themselves (and this information is very doubtful).

We referred to authors in square brackets as we had originally written it in an anonymised fashion for peer review and we failed to rectify this in our editing.  We have now added in this reference.  As we outline below, we build on our existing work because there is otherwise a lack of work in this area to build on. We are one of the very few teams exploring robotics in public management and the only one to our knowledge in relation to policy capacity. We are not reasserting existing knowledge in this article.  We are building on existing literature to develop a new framework of policy capacity that has not been previously published elsewhere.  Therefore we present new conclusions and not ones that we have published before. We would agree that governments can support acceleration of these technologies, but our point is that at present they are not with respect to care robotics in these countries.  If the reviewer could articulate in more detail why these new findings are ‘doubtful’ beyond simply labelling our findings as ‘ridiculous’ but without any critique in engaging with the literature on policy capacity and robotics that would be helpful to us in developing the argument further.  As it stands it is difficult for us to identify what is problematic with our statements.    

Table 1 appears to have been borrowed from [10], as follows from this article. It is hardly appropriate to completely reproduce the table from someone else's article.

It is quite usual to build on a framework set out in a previous study.  We have referenced this table and have further sought to build on the general set of skills and abilities outlined by Wellstead and Steadman in relation to this specific area. 

The whole article is not scientific, it's just a selection of some interviews, perhaps from random people. But even if the interviews were taken from professionals, the answers show that there is nothing valuable and new in these interviews. The results of the interview are not even processed properly, all the meaningful information in the conclusion consists in the same dubious phrase that was previously unreasonably given in the introduction, and about which the authors themselves write that they had said it before.

This is a qualitative study that set out to explore the perspectives of a range of stakeholders working in or with Australian on New Zealand governments.  While the sampling approach was purposive, it was not a set of “random people” but as outlined in the methodology we sought to speak to a mixture of those working in government agencies, suppliers of technologies or those who are actively involved in studying these technologies and their implementation within Australian and New Zealand government-funded agencies (see Table 2).  The interview transcripts were coded according to the processes set out in the paper.  If the reviewer could highlight which particular aspects of the description of the methodology are problematic, we would be happy to respond to this.  None of the other reviewers have indicated a problem with the methodology. We believe there is an original contribution to knowledge as we go beyond what has previously been published and have set out the policy capacity required to oversee these technologies.  This is not something that we have ever published before and to our knowledge (and careful study of the literature) is not something that another team has published either. 

The article does not contain anything useful new, informative.

It is impossible to improve an article by editing.

I recommend that you unequivocally reject the article.

These comments stand in contrast with those of the other reviewers and without further specificity as to the perceived problems it is rather tricky for the author team to respond to these blanket statements. 

Round 2

Reviewer 4 Report

After correction, the article can be published in this form, additional corrections are not required.